# Discovery of Novel HSP27 Inhibitors as Prospective Anti-Cancer Agents Utilizing Computer-Assisted Therapeutic Discovery Approaches

**DOI:** 10.3390/cells11152412

**Published:** 2022-08-04

**Authors:** Haruna Isiyaku Umar, Adeola Temitayo Ajayi, Nobendu Mukerjee, Abdullahi Tunde Aborode, Mohammad Mehedi Hasan, Swastika Maitra, Ridwan O. Bello, Hafsat O. Alabere, Afees A. Sanusi, Olamide O. Awolaja, Mohammed M. Alshehri, Prosper O. Chukwuemeka, Nada H. Aljarba, Saad Alkahtani, Sumira Malik, Athanasios Alexiou, Arabinda Ghosh, Md. Habibur Rahman

**Affiliations:** 1Department of Biochemistry, Federal University of Technology, Akure PMB 704, Nigeria; 2Computer-Aided Therapeutic Discovery and Design Group, FUTA, Akure PMB 704, Nigeria; 3Department of Microbiology, Ramakrishna Mission Vivekananda Centenary College, Kolkata 700118, India; 4Department of Health Sciences, Novel Global Community Educational Foundation, Hebersham, NSW 2770, Australia; 5Healthy Africans Platform, Research and Development, Ibadan 200001, Nigeria; 6Department of Biochemistry and Molecular Biology, Faculty of Life Science, Mawlana Bhashani Science and Technology University, Tangail 1902, Bangladesh; 7Department of Microbiology, Adamas University, Kolkata 222001, India; 8Department of Biotechnology, Federal University of Technology, Akure 340252, Nigeria; 9Department of Microbiology, Federal University of Technology, Akure 340252, Nigeria; 10Department of Chemistry, Federal University of Technology, Akure 340252, Nigeria; 11Pharmaceutical Care Department, Ministry of National Guard-Health Affairs, Riyadh 11426, Saudi Arabia; 12Department of Biology, College of Science, Princess Nourah bint Abdulrahman University, P.O. Box 84428, Riyadh 11671, Saudi Arabia; 13Department of Zoology, College of Science, King Saud University, P.O. Box 2455, Riyadh 11451, Saudi Arabia; 14Amity Institute of Biotechnology, Amity University Jharkhand, Ranchi 834001, India; 15Department of Science and Engineering, Novel Global Community Educational Foundation, Hebersham, NSW 2770, Australia; 16Microbiology Division, Department of Botany, Gauhati University, Guwahati 781014, India; 17Department of Global Medical Science, Wonju College of Medicine, Yonsei University, Wonju 26426, South Korea

**Keywords:** HSP27 inhibitor, anti-cancer resistance, computer-assisted therapeutic discovery, anti-cancer agent, cancer therapy, molecular docking

## Abstract

Heat shock protein 27 (HSP27) is a protein that works as a chaperone and an antioxidant and is activated by heat shock, environmental stress, and pathophysiological stress. However, HSP27 dysregulation is a characteristic of many human cancers. HSP27 suppresses apoptosis and cytoskeletal reorganization. As a result, it is recognized as a critical therapeutic target for effective cancer therapy. Despite the effectiveness of multiple HSP27 inhibitors in pre-clinical investigations and clinical trials, no HSP27 inhibitor has progressed to the anticancer phase of the development. These difficulties have mostly been attributable to existing anticancer therapies’ inability to target oncogenic HSP27. Highly selective HSP27 inhibitors with higher effective-ness and low toxicity led to the development of combination techniques that include computer-aided assisted therapeutic discovery and design. This study emphasizes the most recent results and roles of HSP27 in cancer and the potential for utilizing an anticancer chemical database to uncover novel compounds to inhibit HSP27.

## 1. Introduction

One of the most abundant small heat-shock proteins (sHSPs) in humans is HSP27 (also known as HSPB1). It is expressed systemically under normal conditions but could be upregulated in cancers, during aging, by oxidative stress and diseases that lead to protein depositions [1,2,3]. HSP27 is a redox-dependent molecular chaperone, that is also non-dependent on ATP, it provides a suitable microenvironment for the thriving of cancerous cells thereby conferring stability to numerous cancer-related genes and proteins that are involved with tumor development [4,5]. In light of this, it prevents cell death by interacting with wide range of apoptotic genes and proteins such as DAXX, ASK1, AKT1, estrogen receptor, cytochrome c, proscaspase-3, TNF-α, Fas, TRAIL, caspase-9, and IKK when phosphorylated [3,4,5,6,7,8]. Phosphorylation is key to HSP27’s activation which leads to the formation of oligomers that in turn facilitate its chaperone’s activity under cellular pressure [4,8]. This process is championed by kinases such as MAPKAP2, p38 MAPK and p90RSK [9]. Consequently, HSP27 plays a vital role in the physiology of cells in numerous disease states, including cancer (Figure 1).

Overexpression of HSP27 had been noticed in many types of human cancers such as breast cancer [4], prostate cancer [14], gastric cancer [6,8], ovarian cancer [6,15], brain cancer- glioblastoma [16], liver cancer [17,18], lung cancer [19], colorectal cancer [20], and pancreatic cancer [21]. This overexpression of HSP27 in cancer promotes progression of cancerous cells via apoptosis inhibition, resistance to treatment and tumorigenicity [4,7]. There-fore, HSP27 will be one of key therapeutic target to be considered in the fight against cancer and treatment resistance in cancer. This stand had been supported by several studies [2,5,8,9,14,18].

Although, HSP27 is a difficult target because it is also an ATP-nondependent molecular chaperone. However, a recent strategy used is to develop HSP27 inhibitors that can inhibit it dimerization, by creating a cross-link between HSP27 proteins [5,22] through the C-terminal, since it is found wanting in inter-dimer interactions that facilitates its oligomer formation [22,23]. A recent study showed that ivermectin inhibits the oligomerization via binding on the N-terminal’s ATP pocket [9]. Both termini are involved in dimerization or oligomerization [22]. Three classes of inhibitors have been developed viz: small molecules, Peptide aptamers and antisense oligonucleotide (OGX-427). 

The small molecules include bromo-vinyl-deoxyuridine (BVDU or brivudine), J2 and quercetin. BVDU show promising outcome in clinical studies but was found to increase the toxic effects of gemcitabine in some individuals [5,21]. J2 is a synthetic chromone molecule that possesses a convincing potent cross-linking activity which is considered as a novel approach to counter HSP27-induced resistance [5]. Peptide aptamers include PA11 and PA50 that binds to HSP27 specifically thereby interfering with dimerization and oligomerization that negatively regulates HSP27 functions [5]. However, there are some hindrances in the use of protein aptamer such as the limit to the size of protein of interest, incapability to use protein complex with membrane and working in an environment free from RNases [5]. Antisense oligonucleotide that targets HSP27 is OGX-427. It targets HSP27′s mRNA that have been tasted against Pancreatic and lung cancer xenografts in combination with chemotherapy [5,24].

Despite the efficacy of these group of HSP27 inhibitors, there are still some toxicity related issues with some having poor efficacy hence, the need to discover, develop or design more compounds that can interfere with HSP27-mediated resistance in cancer. 

Our current study is aimed at discovering a new compound from an anti-cancer compound’s database from Selleck Chemicals (Houston, TX, USA) that can inhibit HSP27 through the use of in-silico techniques. To achieve this, we embark on screening this unique database of anti-cancer compounds using Lipinski’s Rule of five (RO5), number of rotatable bonds, and medicinal chemistry properties. Afterwards, those molecules that scaled through this screening were docked against HSP27 as well screen the resulting hit compounds for in-silico toxicological properties. Finally, the protein-ligand complex stability was studied through molecular dynamics simulation. We found two compounds namely; FR180204 and XAV-939 from the database as promising as anti-HSP27 molecules. However, it is expedient to further this study via wet laboratory evaluations.

## 2. Materials and Methods

### 2.1. Compound Datasets

About 3547 anticancer compounds were outsourced from a compound library of high throughput Server (selleckchem.com; assessed on 1 June 2021). This compound library is made of unique collections of 3864 anticancer compounds for multiple cancers such as breast cancer, leukaemia, lung cancer, lymphoma, cervical cancers, prostate cancer, etc. Most of these compounds have been approved by FDA and have been reported in articles related to tumour research. The control drugs adopted for this current work is J2 (CID: 135384973) and Brividine (CID: 446727).

### 2.2. Screening of Compound Library for Drug Likeness

All the compounds were screened for drug likeness using parameters such as the Lipinski’s rule of five (RO5) [25], number of rotatable bonds (nRot) and the medicinal chemistry properties. This screening experiment was achievable through the SwissADME Server (http://www.swissadme.ch/index.php, assessed on 1 June 2021) [26]. Components such as molecular weight (MW), lipophilicity (AlogP), number of hydrogen bond donors (HBD), and number of hydrogen bond acceptors (HBA) comprised Lipinski’s rule of five, whereas Lead similarity, Brenk, and PAINS alert included medicinal chemistry evaluation. After putting these compounds to a drug-likeness test, those that passed were chosen for further processing, such as molecular docking against HSP27.

### 2.3. Selection and Preparation of Protein Target

Heat-shock protein 27 (HSP27) of Humans (PDB ID: 4MJH) was picked from the Pro-tein Database (PDB) (https://www.pdb.org/pdb, assessed on 1 June 2021) with a resolution of 2.6 [27], which had been previously used [23]. The protein was rendered nascent by removing heteroatoms such as water molecules, ions, and ligands, and then its energy was reduced using the protein preparation and minimization methods in Cresset Flare software, version 4.0 (https://www.cresset-group.com/flare/, assessed on 1 June 2021). Under the General Amber Force Field (GAFF), protein minimization was performed using a gradient cutoff of 0.200 kcal/mol/and 2000 iterations [28].

### 2.4. Preparation of Compounds for Molecular Docking against HSP27

The 3D conformers in the structure data file (SDF) of control drugs and those that scaled through drug likeness screening were obtained from PubChem, a chemical repository linked to the National Centre for Biotechnological Information (NCBI), one of the world’s largest collections of freely accessible chemical information. Furthermore, using the program Open babel within Python Prescription, the resulting compounds’ structures were transformed into their most energetic and stable conformations using Merck molecular force field (MMFF94) (version 0.8) [26].

### 2.5. Molecular Docking of Compounds against HSP27

The molecular docking was achieved using AutoDock Vina in open-source Python Pre-scription 0.8 [29] to acquire possible poses or orientations and binding energy (BE) of com-pounds at the binding site of HSP27. A target region for HSP27 analogous to the binding region of other monomers of HSP27 was attuned with the aid of the grid box with dimensions (25.1378 × 20.3355 × 20.8123) Å, and the center was adjusted based on the site of monomer binding in HSP27 comprising of Arg140, Thr139, Phe138, Cys137, Arg136, Phe104, His103, Val101, Asp100, Gln44, Phe33, Tyr23, Glu119 and Cys141 [22,23,27,30].

After the docking run, compounds with docking score (binding energy) below that of the control drugs were subjected to molecular visualization in order to analyze their molecular interaction fingerprints through PyMOL© Molecular Graphics (version 2.4, 2016, Schrödinger LLC, New York, NY, USA) [31] and BIOVIA’s Discovery studio (version 2016). These molecular interaction finger-prints to be analyzed for each protein-ligand complex are the hydrogen bonds, hydrophobic interactions, and electrostatic linkages between amino acids of the protein and atoms of the ligands.

### 2.6. ADMET Prediction In Silico

ADMET (Absorption, Distribution, Metabolism, Excretion, and Toxicity) is essential in the early stages of the drug discovery and design pipeline for analyzing the pharmacokinetics of potential drug candidates. After molecular docking studies, the ADMETSar server was used to predict the ADMET characteristics of the compounds with the highest number of hits [32,33]. The server was fed with the SMILE Strings of the compounds from PubChem (https://pubchem.ncbi.nlm.nih.gov/compound/, assessed on 1 June 2021) through it search bar and ran for prediction of ADMET properties.

### 2.7. Molecular Dynamics Simulations

The stability and dynamism of three complexes of BVDU, FR180204 and XAV-939 from molecular docking studies that exhibited desirable pharmacokinetic profile was executed by GROMACS 2019.2 package [34]. Using the Gromos43a1 force field, the topologies for the apo form of the protein and protein-ligand complexes were generated [35]. After constructing the topology file, all complexes and single protein structures were placed in a cubic box, solvated with explicit simple point charge (SPC) water model, and neutralized with ions from 0.15 M NaCl.

In addition, these structures were made more relaxed by applying an approach for energy reduction that incorporated both the steepest descent method and the Verlet cut-off scheme. The energy cost per molecule was set at 10 kJ/mol for a total of 50,000 cycles using this technique. During the first hundred picoseconds of the trajectory, the protein and ligands complex underwent an equilibration step on both NVT (constant volume) and NPT (constant pressure). The simulation analysis for a 100-nanosecond run was computed using a temperature of 300 kelvins, a pressure of 1 atmosphere, and a time step of 2 femtoseconds following the equilibration phase.

Afterwards, the visualization of trajectory files produced herein were carried out with the aid of Qtgrace software (ver. 0.2.6) to check the deviation of each protein and complex so as to establish the system’s stability in a water-contained environment. In order to study the deviation between protein and Ligand complexes; Root mean square deviation (RMSD), Root mean square fluctuation (RMSF), Radius of gyration (Rg), Hydrogen bonds, and Solvent accessible surface area (SASA) were explored.

## 3. Results

### 3.1. Drug Likeness Screening 

The 3547 compounds sourced from selleckchem database (www.selleckchem.com, assessed on 1 June 2021) were subjected to Lipinski’s rule of five and number of rotatable bonds. The idea was to select compounds that have molecular weight between 300–450, AlogP between 2.5–4.0, HBD between 0–5, HBA between 0–9 and nRot between 2–5. 261 compounds were found to pass the screening. Furthermore, medicinal chemistry parameters were used to screen the 261 compounds. Here, we checked for those compounds that do not violate lead likeness, PAINS and Brenk’s alerts. This yielded 47 compounds (Figure 2) with their molecular weight falling between 300–360 g/mol (Table 1). The significance of this step is to obtain compounds whose chance of being orally active is high.

### 3.2. Docking-Derived Binding Affinity

Out of the 47 compounds from the database, 13 compounds showed they are better binders (between −7.0 and −8.2 kcal/mol) to HSP27 than the two control drugs (BVDU = −6.0 kcal/mol and J2 = −5.8 kcal/mol) selected in this current in-silico study (Table 2). APY29 (−7.7 kcal/mol), FR180208 (−8.1 kcal/mol) and Fluorescein (−8.2 kcal/mol) were better binders than the 14 compounds. However, fluorescein is a dye so was not pursued as pharmacologically active drug, while APY29 has some infarctions in respect to ADMET profiling (Table 3). The molecular interaction framework of Control drug (BVDU) with better binding energy, FR180204 and XAV-939 is presented in Figure 3. There was formation of two hydrogen bonds with His103 by FR180204 with distances of 3.72 Å and 3.44 Å; while atoms of XAV could be seen to interact with Cys137 (at distances of 4.48 Å and 4.65 Å) and Arg140 (at distances of 3.72 Å and 3.72 Å).

BVDU was observed to establish contact by hydrogen bonding not only with His103 and Arg140 but with Asp100 and Val101. Another interesting observation is the amino acids involved in hydrophobic interactions with our leads and control. Phe104, Cys137, Val85, Leu99, Val101, Arg140, Asn102, Arg136 and Ile134 were involved in this interaction. Other interactions observed were Pi-Stacking and halogen bond linkage. The 2D molecular interaction with amino acid residues and the remaining 11 compounds is presented in Figure 4.

### 3.3. ADMET Prediction 

The ADMET prediction was carried out to ascertain how the 13 hit compounds are absorbed, distributed, metabolized, eliminated and how toxic they can be. The outcome of the evaluation is presented in Table 3. FR180204 and XAV-939 showed good ADMET profile. Both have no tendency to cause mutation against *Salmonella typhinurium* (Ames mutagenesis) and also were observed to be non-inhibitor of human either-a-go-go gene (herG).

Similarly, both compounds showed ability to cross the blood-brain-barrier (BBB), absorbed through caco2 and good intestinal absorption. FR180204 may serve as substrate P-glycoprotein both compounds are not inhibitors of P-glycoprotein. FR180204 was found to be substrate to cytochrome P450 isoform 3A4 but not with other isoforms present, while XAV-939 was found to be non-substrates for the cytochrome P450 isoforms.

### 3.4. Molecular Dynamics Simulation of HSP27-Drug Complexes

After the ADMET profiling, the association of HSP27-compound complexes were ex-amined and the dynamic stability of screened compounds was studied using MD simulation at 100 ns. This was achieved with the aid of GROMACS 2019. Analyzing the RMSD (Figure 5A), HSP27-BVDU was found to reach 0.4 nm at 5 ns until 30 ns where it dropped to 0.2 nm for al-most throughout the remaining MD run. Around the same run time we found that HSP27-FR180204 complex had a value of 0.38 nm which increased to 0.46 nm at 35 ns which it maintains throughout the run with some slight fluctuations along the production run. We observed that HSP27-XAV-939 complex reached a peak RMSD of 0.4 nm at 18 ns followed by fluctuations. Our results herein suggest that the binding of FR180204 and XAV-939 to HSP27 may prompt conformational alterations.

Consistent with this, the analysis of RMSF against HSP27 residue number showed that FR180204 complex showed higher oscillations in backbone residues when compared to BVDU and XAV-939 systems (Figure 5B). Another key parameter evaluated is the radius of gyration (Rg) to ascertain the compactness variations of a protein-ligand complex. The control drug, BVDU showed a high Rg fluctuation compared to other compounds (Figure 5C). This is consistent with the docking results of BVDU that showed the highest binding free energy of −6.0 kcal/mol with XAV-939 and FR180204 showed the least binding free energy of −7.3 and −8.1 kcal/mol (Table 2).

The Solvent accessibility surface area (SASA) of the complexes was analyzed along the simulation period (Figure 5D). HSP27-BVDU complex showed at the beginning a SASA of 62 nm^2^ but as the run progresses it dived down to 50 nm^2^. Similarly, complexes of XAV-939 and FR180204 had almost same SASA value at the beginning of the run as BVDU but their final SASA was found to be higher than that of BVDU. During the simulation, the formation and stability of H-bonds were looked at (Figure 5E). Because H-bonds are so important to the specificity, metabolism, and absorption of drugs, they are a key part of designing and finding new drugs. From Figure 5, the results showed that BDVU and XAV-939 could each make one hydrogen bond with the amino acid residues Lys 141 and Asp100, while FR180204 could not make a hydrogen bond but did make a pi-stacking interaction with the amino acid residues His124 and Phe138.

Even though the MD simulation is still going on, FR180204 was able to form four hy-drogen bonds at 40 ns (Figure 5E). So, H-bonds between active site residues and complexes kept them stable. This was the case with XAV-939 (Figure 6). The docking and MD simulation analyses showed that FR180204 and XAV-939 were more likely to bind to HSP27 than our control drug, BVDU. But FR180204 had a high docking score (−8.1 kcal/mol) and was able to form pi-stacking interactions with two of the essential amino ac-ids of the HSP27 binding domain. 

## 4. Discussion

The roles of HSP27 in cancer has been well documented. It plays a key role in anti-cancer treatment resistance, inhibition of apoptosis and tumor progression. This make HSP27 a suitable drug target to combat cancer in the world today. We screened a database containing 3547 anticancer compounds as at the time of this work, for potential to bind and interact with key amino acid residues of HSP27.

First, these compounds were screened for drug-likeness qualities using Lipinski’s rule of five, the number of rotatable bonds, and Lilly’s medicinal chemistry filter, which includes parameters like lead likeness and PAINS (Pan assay interference compounds), and Brenk’s alert. The rule of 5 indicates that when the number of H-bond donors exceeds 5, the number of H-bond acceptors surpasses 10, the MW exceeds 500, and the Lipophilicity exceeds 5 [22]. The rationale for performing this step early in this current in-silico work was to exclude compounds with properties that are not compatible with an acceptable pharmacokinetic profile. Most of these molecules may fail at the penultimate phase of clinical trials and later fail during drug discovery.

The number of rotatable bonds of a drug-like molecule should not exceed 10 as the higher this value, the flexible the molecule which could affects its oral bioavailability [26]. The medicinal chemistry filters were implemented in order to screen out compounds that have problematic fragments linked to target promiscuity, putative toxicity, that are chemo-reactive, will be unstable during metabolism, or possessing features responsible for blameworthy pharmacokinetics behaviors. Therefore, compounds that scaled through these filtering process can be tagged drug like. In this in-silico study, 47 anti-cancer compounds (Figure 2) were found to scale through the drug like filters we used.

The 47 anti-cancer compounds, J2 and BVDU were subjected to molecular modelling steps that is deployed to predict how the compounds interacts with HSP27. This step is termed molecular docking and it is employed to define the binding attraction of a lead compound and its most probable binding configuration [36]. 13 compounds had a binding score better than the two control drugs (J2 and BVDU) selected for this study (Table 2). From Figure 3A, we could note that our lead compounds occupy similar space on the binding region of the target protein when analyzing their binding modes. Furthermore, we note that the interactions of these lead compounds with the amino acid residues found in the active site of HSP27 are similar with those reported in earlier studies [23,30,37]. They reported that Arg136, Phe104, His103, Val101, Cys137, Thr139, Phe138, Arg140 and Asp100 were crucial in the binding region of HSP27.

The goal of screening all 13 compounds through the ADMETSar online server was to defeat the challenge of toxicity/side effects earlier on in the drug design and discovery voyage prior to wet lab experimentation. So, identifying lead molecules with better pharmacokinetic activity without side effects is most pertinent. Parameters such as absorption, distribution, metabolism and toxicity of the 13 compounds were predicted using a computer-based technique since the standard laboratory experiment for this evaluation are cost effective and time consuming. Using these techniques, we discovered FR180204 and XAV-939 as the top two hit compounds (Table 3). FR180204 is a selective target of the extracellular signal-regulated kinase 1 and 2 (ERK 1/2) pathway in the Mitogen-activated protein kinase (MAPK). It inhibits ERK through competition with ATP (Type I ATP inhibitor) [38,39]. FR180204 has been shown to have an antiproliferative effect in different cancer cell lines, including adrenocortical, mesothelioma, prostate, pancreatic, and colorectal cell lines [40,41]. Similarly, XAV-939 has been shown to target the Wnt/B-catenin signaling pathway by inhibiting tankyrases 1 and 2. It has been demonstrated that targeting the Wnt/B-catenin signaling pathway is effective in many cancers that includes breast cancer, colorectal cancer and WTK1 human lymphoblastoid cells [42].

A molecular dynamics simulation was deployed to provide insights about the interaction, stability of two hit compounds and the control drug inside HSP27, and conformational derivations of these compounds or HSP27 that may occur inside the biological system in various conditions with respect to time [38]. From this present study, the two hit compounds showed attributes that are in line with stability cum affinity to HSP27 (Figure 5 and Figure 6). Consequently, making both hit compounds potential binders and inhibitors of HSP27. However, an immediate call for follow up in vitro and in vivo experiments to validate their activity against HSP27. The outcome of our present study showed that XAV-939 and FR180204 have the potential to bind and consequently inhibit the activity of HSP27 in anti-cancer drug resistance.

## 5. Conclusions

We used a virtual drug discovery approach to explore the possibility of discovering new com-pounds that have inhibitory potential against HSP27 in-silico from a database of anti-cancer compounds. By utilizing Lipinski’s Rule of Five (RO5), the number of rotatable bonds, and medicinal chemistry properties, we are able to screen a unique database of anticancer compounds. At the end of our investigation, we identified that the compounds entitled as XAV-939 and FR180204 acted as active inhibitors of HSP27’s. Molecular dynamics simulation of these com-pounds in complexes with HSP27 over 100 ns revealed the ability to stabilize the protein. Also, these compounds were found to show good pharmacokinetic profile, indicating a safe treatment option in the management of human cancers. Consequently, we can foretell that these com-pounds hold good candidacy to be established for the treatment of human cancer.

## Figures and Tables

**Figure 1 cells-11-02412-f001:**
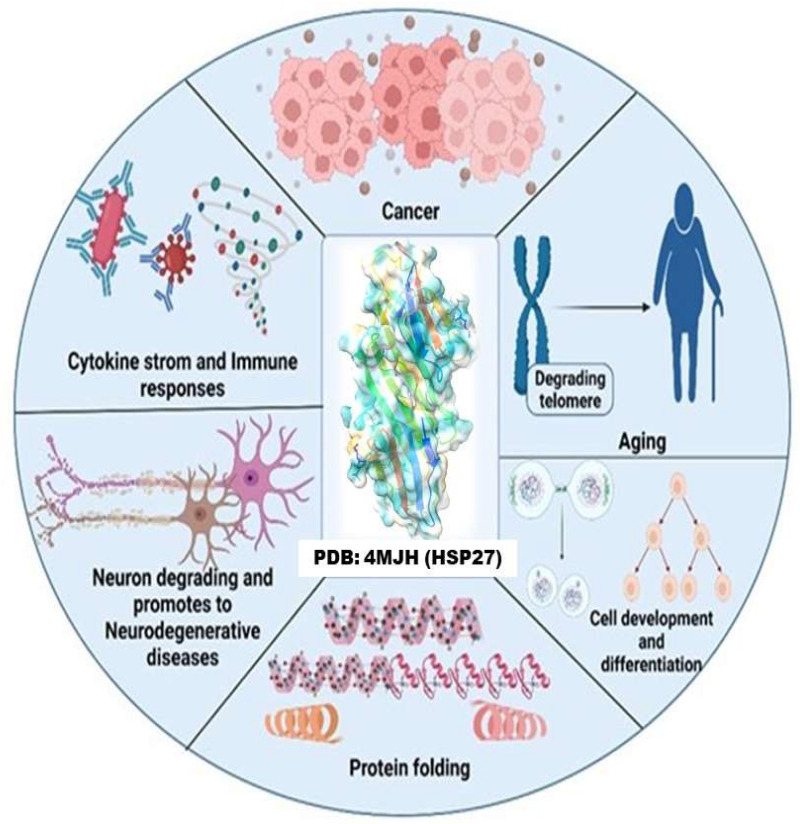
By chaperone activity, HSP27 regulates protein folding, regulates immune response, promotes cancer progression, induces resistance to anticancer therapeutics, increases aging, aggravation of neurodegenerative diseases, and differentiates cells [10,11,12,13].

**Figure 2 cells-11-02412-f002:**
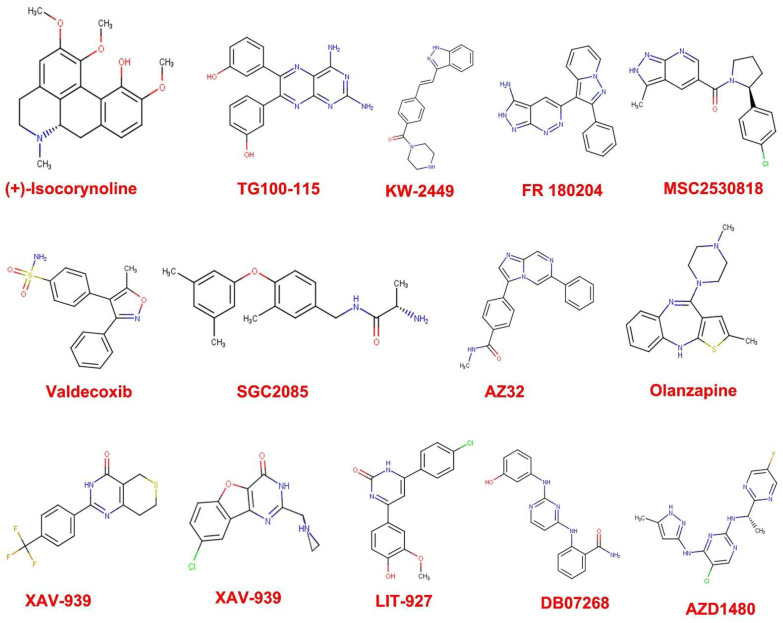
Chemical structures of druglike compounds from sellekchem chemical repository for anti-cancer compounds in 2D.

**Figure 3 cells-11-02412-f003:**
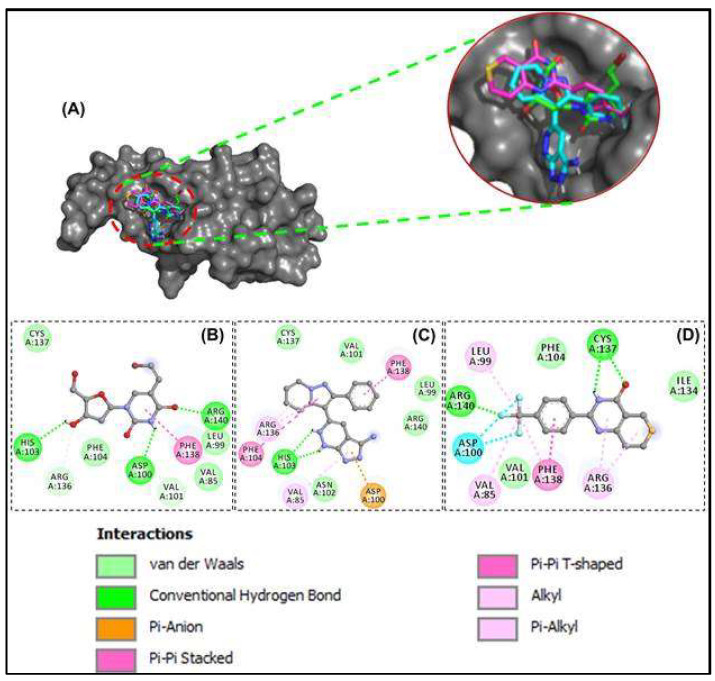
Molecular docking of anti-cancer compounds against HSP27. (**A**) The 3D binding configuration of BVDU (green), FR180204 (cyan) and XAV-939 (pink). All three ligands occupied similar spot of our protein target. The molecular interaction fingerprints in 2D showed that atoms of our ligands (**B**) BVDU (**C**) FR180204 (**D**) XAV-939 interacted with amino acid residues important for binding between two HSP27 monomers.

**Figure 4 cells-11-02412-f004:**
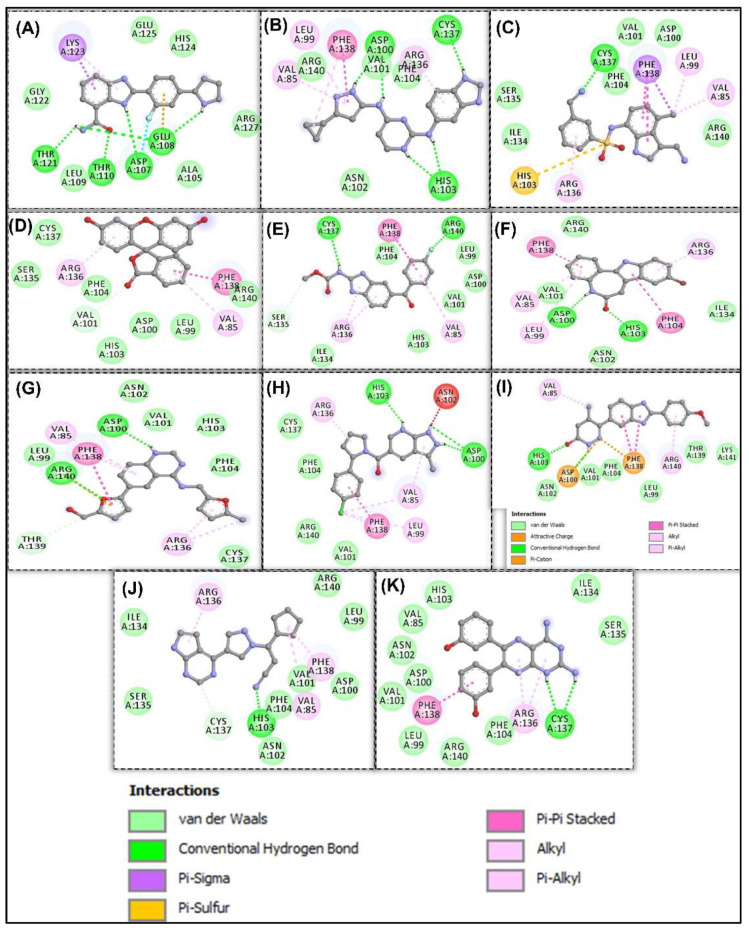
Molecular interaction of anti-cancer compounds docked against HSP27. 2D of anti-cancer compounds (**A**) A-966492 (**B**) APY29 (**C**) E7820 (**D**) Fluorescein (**E**) Flubendazole (**F**) Kenpaulline (**G**) ML167 (**H**) MSC2530818 (**I**) Pimobendan (**J**) S-Ruxolitinib (**K**) TG100-115 interact-ing with amino acid residues of HSP27.

**Figure 5 cells-11-02412-f005:**
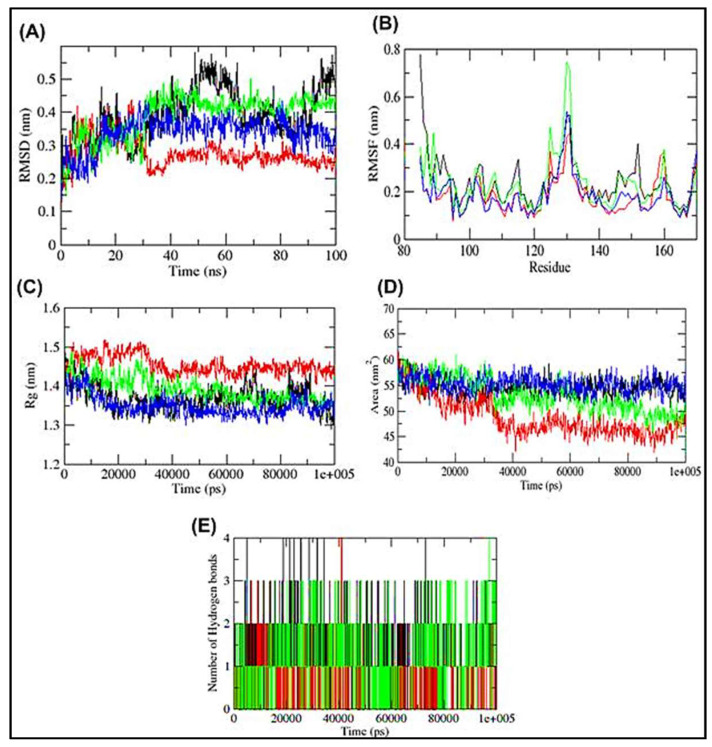
Molecular Dynamics Simulation of HSP27 Apo protein (black), BVDU (red), FR180204 (green) and XAV-939 (blue) over a simulation run of 100 ns. (**A**) RMSD (**B**) RMSF (**C**) Radius of gyration (Rg) (**D**) Solvent Accessible Surface Area (SASA) (**E**) Number of Hydrogen bonds during the MDS run of BVDU (black), FR180204 (red) and XAV-939 (green).

**Figure 6 cells-11-02412-f006:**
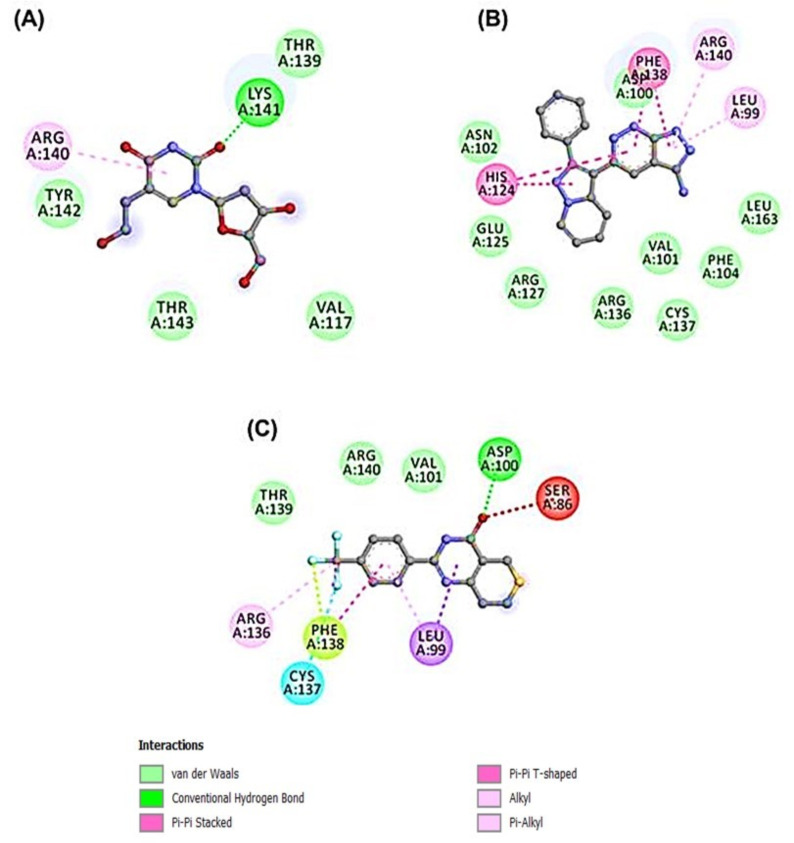
Molecular Interactions obtained from Molecular Dynamics Simulation of three complexes for a period of 100 ns. (**A**) BVDU (**B**) FR180204 (**C**) XAV-939.

**Table 1 cells-11-02412-t001:** Screened Molecules from selleckchemical’s unique library of anti-cancer compounds.

S/N	Molecules	PID	MW	PAINS Alert	Brenk Alert	Lead Likeness Violations
**1**	A-966492	16666333	324.35	0	0	0
**2**	A2AR antagonist 1	53466958	309.3	0	0	0
**3**	AG-14361	9840076	320.39	0	0	0
**4**	APY29	42627755	332.36	0	0	0
**5**	ASP-9521	25210792	330.42	0	0	0
**6**	AZD1480	16659841	348.77	0	0	0
**7**	AZ32	134814488	328.37	0	0	0
**8**	Broxyquinoline	2453	302.95	0	0	0
**9**	CCT128930	17751819	341.84	0	0	0
**10**	DB07268	16058637	321.33	0	0	0
**11**	Eupatilin	5273755	344.32	0	0	0
**12**	E7820	196970	336.37	0	0	0
**13**	Fluorescein	16850	332.31	0	0	0
**14**	Flubendazole	35802	313.28	0	0	0
**15**	FR 180204	11493598	327.34	0	0	0
**16**	GNE-0877	69093374	339.32	0	0	0
**17**	Hydroquinidine	91503	326.43	0	0	0
**18**	(+)-Isocorynoline	10143	341.4	0	0	0
**19**	Kenpaullone	3820	327.18	0	0	0
**20**	KW-2449	11427553	332.4	0	0	0
**21**	Lificiguat (YC-1)	5712	304.34	0	0	0
**22**	LIT-927	137287575	328.75	0	0	0
**23**	Longdaysin	49830252	335.33	0	0	0
**24**	ML167	44968231	335.36	0	0	0
**25**	MSC2530818	118879529	340.81	0	0	0
**26**	Niraparib (MK-4827)	24958200	320.39	0	0	0
**27**	Nocodazole	4122	301.32	0	0	0
**28**	Olanzapine	135398745	312.43	0	0	0
**29**	Omeprazole	4594	345.42	0	0	0
**30**	Oxfendazole	40854	315.35	0	0	0
**31**	Pimobendan	4823	334.37	0	0	0
**32**	PNU 282987	9795278	300.2	0	0	0
**33**	PI-103	9884685	348.36	0	0	0
**34**	PP2	4878	301.77	0	0	0
**35**	PP121	24905142	319.36	0	0	0
**36**	R112	9904854	312.3	0	0	0
**37**	Ruxolitinib (INCB018424)	25126798	306.37	0	0	0
**38**	SCH58261	176408	345.36	0	0	0
**39**	SGC2085	121231417	312.41	0	0	0
**40**	S-Ruxolitinib (INCB018424)	50878566	306.37	0	0	0
**41**	Tenatoprazole	636411	346.4	0	0	0
**42**	Torkinib (PP242)	135565635	308.34	0	0	0
**43**	TG100-115	10427712	346.34	0	0	0
**44**	Valdecoxib	119607	314.36	0	0	0
**45**	XL413 (BMS-863233)	135564632	325.17	0	0	0
**46**	XAV-939	135418940	312.31	0	0	0
**47**	ZM241385	176407	337.34	0	0	0

PID = PubChem Identity; MW = Molecular weight; PAINS = Pan-assay interference compounds.

**Table 2 cells-11-02412-t002:** The docking-derived binding affinity of anticancer compounds with good drug-like properties against HSP27.

S/N	Ligand	Binding Energy (kcal/mol)
1.	A-966492	−7.2
2.	A2AR antagonist 1	−5.4
3.	AG-14361	−6.4
4.	APY29	−7.7
5.	ASP-9521	−5.6
6.	AZD1480	−6.2
7.	AZ32	−6.8
8.	Brivudine (BVDU)	−6.0
9.	Broxyquinoline	−5.3
10.	CCT128930	−6.0
11.	DB07268	−6.9
12.	Eupatilin	−6.4
13.	E7820	−7.1
14.	Fluorescein	−8.2
15.	Flubendazole	−7.3
16.	FR 180204	−8.1
17.	GNE-0877	−6.2
18.	Hydroquinidine	−6.1
19.	(+)-Isocorynoline	−6.4
20.	J2	−5.8
21.	Kenpaullone	−7.5
22.	KW-2449	−6.9
23.	Lificiguat (YC-1)	−6.5
24.	LIT-927	−6.6
25.	Longdaysin	−6.5
26.	ML167	−7.3
27.	MSC2530818	−7.2
28.	Niraparib (MK-4827)	−6.7
29.	Nocodazole	−6.3
30.	Olanzapine	−5.9
31.	Omeprazole	−6.0
32.	Oxfendazole	−6.5
33.	Pimobendan	−7.2
34.	PNU 282987	−5.8
35.	PI-103	−6.7
36.	PP2	−6.4
37.	PP121	−6.7
38.	R112	−6.8
39.	Ruxolitinib (INCB018424)	−6.9
40.	SCH58261	−6.4
41.	SGC2085	−6.7
42.	S-Ruxolitinib (INCB018424)	−7.2
43.	Tenatoprazole	−5.8
44.	Torkinib (PP242)	−6.9
45.	TG100-115	−7.6
46.	Valdecoxib	−6.8
47.	XL413 (BMS-863233)	−6.9
48.	XAV-939	−7.3
49.	ZM241385	−6.7

**Table 3 cells-11-02412-t003:** In-silico ADMET profiling of hit compounds using ADMETSar online server.

ADMET PROFILES	Kenpaullone	Pimobendan	Fluorescein	Flubendazole	E7820	TG100-115	FR180204
Ames mutagenesis	+	+	-	-	+	-	-
Blood Brain Barrier	+	+	-	+	+	+	+
Caco-2	+	-	-	+	-	-	+
CYP1A2 inhibition	+	+	-	+	+	+	+
CYP2C19 inhibition	-	+	-	-	+	+	+
CYP2C9 inhibition	-	-	+	-	-	+	-
CYP2C9 substrate	-	-	-	-	+	-	-
CYP2D6 inhibition	+	-	-	-	-	-	-
CYP2D6 substrate	-	-	-	-	-	-	-
CYP3A4 inhibition	+	+	+	-	+	-	-
CYP3A4 substrate	+	+	+	+	+	-	+
Human either-a-go-go inhibition	-	+	-	+	-	-	-
OCT2 inhibitior	-	-	-	-	-	-	-
Human Intestinal Absorption	+	+	+	+	+	+	+
Human oral bioavailability	+	+	+	-	+	+	+
P-glycoprotein inhibitior	-	-	-	-	-	-	-
P-glycoprotein substrate	-	-	-	-	-	-	+
Subcellular localization	Mitochondria	Mitochondria	Mitochondria	Mitochondria	Lysosomes	Mitochondria	Mitochondria
**ADMET PROFILES**	**A-966492**	**APY29**	**ML167**	**S-Ruxolitinib**	**MSC2530818**	**XAV-939**	**BVDU**
Ames mutagenesis	-	-	-	-	-	-	-
Blood Brain Barrier	+	+	+	+	+	+	+
Caco-2	-	-	-	-	+	+	-
CYP1A2 inhibition	+	+	+	+	-	+	-
CYP2C19 inhibition	+	-	+	-	+	-	-
CYP2C9 inhibition	+	-	+	-	+	-	-
CYP2C9 substrate	-	-	-	-	+	-	-
CYP2D6 inhibition	-	-	-	-	-	-	-
CYP2D6 substrate	-	-	-	-	-	-	-
CYP3A4 inhibition	-	-	-	-	-	-	-
CYP3A4 substrate	+	+	+	+	+	-	-
Human either-a-go-go inhibition	-	+	+	+	-	-	-
OCT2 inhibitor	+	+	-	+	-	+	-
Human Intestinal Absorption	+	+	+	+	+	+	-
Human oral bioavailability	+	+	-	+	-	-	+
P-glycoprotein inhibitor	+	-	+	-	-	-	-
P-glycoprotein substrate	+	+	-	+	-	-	-
Subcellular localization	Mitochondria	Mitochondria	Mitochondria	Mitochondria	Mitochondria	Mitochondria	Nucleus

## Data Availability

Not applicable.

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
