# Peer review of "Discovery of Novel HSP27 Inhibitors as Prospective Anti-Cancer Agents Utilizing Computer-Assisted Therapeutic Discovery Approaches"

_cells, 2022, doi:10.3390/cells11152412_

Round 1
Reviewer 1 Report
This manuscript sounds very interesting but it seems to be modestly detailed.
My comments as follows:
Introduction- please move the sentence “Consequently, 86 HSP27 plays a vital role in the physiology of cells in numerous disease states, including 87 cancer (Figure 1)” and the related figure 1 after this “Phosphorylation is key to HSP27’s activation which leads to the 64 formation of oligomers that in turn facilitate its chaperone’s activity under cellular pres- 65 sure [4], [8]” and before of lined 67-86.
Figure 1- please modify PdB in PDB
The three classes of HSP27 inhibitors should be accompanied by a proper scheme in order to help the reader towards rapid and deep comprehension of the chemical space so far exploited in this context.
The last sentence should be accompanied by a brief sum up of the obtained results and future perspective of this study.
Please, pay attention to reference style and citation order, ref. 29 and 30 are not sequentially cited along the manuscript.
Experimental section should be described more in details. Partial charge method to ligands should be added, while it seems to be unclear how the Authors identify the putative protein site for the reported docking calculation. Also the number of retained compounds for MD simulations should be added (both in the experimental and results sections).
Quality of figure 2 seems to be poor, please revise it dividing the compounds based on the main chemical core.
Table 2 should be accompanied by a proper caption and by the scoring function unit of measure. Explain why some compound are in bold style. The same for Table 3.
Experimental validation at least of one of the selected compound should be added, it should be noticed that virtual screening was performed based on the 3D-structure of an HSP27 fragment. It seems to be hard relying on it without final biological assays.
Author Response
#Reviwer-1:
- Introduction- please move the sentence “Consequently, 86 HSP27 plays a vital role in the physiology of cells in numerous disease states, including 87 cancer (Figure 1)” and the related figure 1 after this “Phosphorylation is key to HSP27’s activation which leads to the 64 formation of oligomers that in turn facilitate its chaperone’s activity under cellular pres- 65 sure [4], [8]” and before of lined 67-86.
Answer: The authors appreciate your comment. The point raised have been attended to accordingly.
- Figure 1- please modify PdB in PDB
Answer: Thank you for the comments. We have edited in the revised manuscript.
- The three classes of HSP27 inhibitors should be accompanied by a proper scheme in order to help the reader towards rapid and deep comprehension of the chemical space so far exploited in this context.
Answer: The authors appreciate your comment. The point raised have been attended to accordingly.
- The last sentence should be accompanied by a brief sum up of the obtained results and future perspective of this study.
Answer: Thank you the comments. It has been addressed accordingly.
- Please, pay attention to reference style and citation order, ref. 29 and 30 are not sequentially cited along the manuscript.
Answer: Referencing style and order has been corrected.
- Experimental section should be described more in details. Partial charge method to ligands should be added, while it seems to be unclear how the Authors identify the putative protein site for the reported docking calculation. Also, the number of retained compounds for MD simulations should be added (both in the experimental and results sections).
Answer: The authors appreciate your comment. The point raised have been attended to accordingly.
- Quality of figure 2 seems to be poor, please revise it dividing the compounds based on the main chemical core.
Answer: Thank you the comments. It has been addressed accordingly.
- Table 2 should be accompanied by a proper caption and by the scoring function unit of measure. Explain why some compound are in bold style. The same for Table 3.
Answer: The authors appreciate your comment. The point raised have been attended to accordingly.
- Experimental validation at least of one of the selected compounds should be added, it should be noticed that virtual screening was performed based on the 3D-structure of an HSP27 fragment. It seems to be hard relying on it without final biological assays.
Answer: The authors appreciate the comments. The authors recommend further studies which include validation via wet lab procedures as this research is a computational work.
Reviewer 2 Report
The article entitled “Discovery of novel HSP27 inhibitors as prospective anti-cancer agents utilizing computer-assisted therapeutic discovery approaches” aims to show the most recent results and roles of HSP27 in cancer emphasizing the potential for utilizing an anticancer chemical database to uncover novel compounds to inhibit HSP27. This was made possible by the use of in silico techniques.
Although the topic is very interesting and opens up many possibilities for the advancement of molecular research on HSP27, the work is often confusing and unclear. For this reason, major improvements are needed.
1) First of all, the references do not respect the formatting required by the journal. Authors are requested to read the rules on the site.
2) On the line 172, the sentence “constant pressure” is repeated twice.
3) On the line 198 in the table caption, add the abbreviations. For example, MW=molecular weight.
4) On line 222, add in the caption of figure 4, what B, C and D represent.
5) On the line 228 in the table caption, why some ligand are in bold?
6) Table 3 must be inserted when quoting in the text. Particularly on the line 205
7) Table 3 shows 14 compounds rather than 13 as indicated on line 230.
8) On line 311, the authors report 47 anti-cancer compounds, but 48 are shown in figure 2 and 49 in the table 2.
9) BVDU is not included as a control in table 2, as the authors say in the line 317.
10) Add references on the line 323.
11) In the discussion, add some bibliographic information on XAV-939.
12) TG 100-115 in Table 3 seems equally promising, in fact several scientific papers report its use in breast, liver and pancreatic cancer. I advise the authors to bring into the discussion the possible presence of molecules that are just as good as those chosen but excluded for a particular reason.
13) Why compound J2 is not present in Figures 5 and 6 as BVDU? On line 108, the authors specify that for this work J2 and BVUD are used as control drugs.
14) Finally, I strongly advise the authors to recreate the tables and images following an alphabetical order of the components examined in order to facilitate the reader in searching and learning the text.
Author Response
#Reviewer-2
The article entitled “Discovery of novel HSP27 inhibitors as prospective anti-cancer agents utilizing computer-assisted therapeutic discovery approaches” aims to show the most recent results and roles of HSP27 in cancer emphasizing the potential for utilizing an anticancer chemical database to uncover novel compounds to inhibit HSP27. This was made possible by the use of in silico techniques. Although the topic is very interesting and opens up many possibilities for the advancement of molecular research on HSP27, the work is often confusing and unclear. For this reason, major improvements are needed.
1) First of all, the references do not respect the formatting required by the journal. Authors are requested to read the rules on the site.
Answer: Thank you the comments. It has been addressed accordingly.
2) On the line 172, the sentence “constant pressure” is repeated twice.
Answer: Thank you the comments. We have edited in the revised manuscript.
3) On the line 198 in the table caption, add the abbreviations. For example, MW=molecular weight.
Answer: We appreciate the comments. We have edited in the revised manuscript.
4) On line 222, add in the caption of figure 4, what B, C and D represent.
Answer: We appreciate the comments. It represents the 2D interactions of the anti-cancer compounds, which have been accurately included in the caption.
5) On the line 228 in the table caption, why some ligand is in bold?
Answer: We appreciate the comments. It represents the 2D interactions of the anti-cancer compounds, which have been accurately included in the caption.
6) Table 3 must be inserted when quoting in the text. Particularly on the line 205
Answer: Thank you the comments. It has been addressed accordingly.
7) Table 3 shows 14 compounds rather than 13 as indicated on line 230.
Answer: Thank you the comments. The authors have corrected the error in the main text and it will be 14 compounds both in main text and table 3.
8) On line 311, the authors report 47 anti-cancer compounds, but 48 are shown in figure 2 and 49 in the table 2.
Answer: We appreciate the comments. Figure 2 and Table 2 have been adjusted to consist 49 compounds i.e. 47 anticancer compounds from our database of choice and two controls J2 and BVDU (brivudine).
9) BVDU is not included as a control in table 2, as the authors say in the line 317.
Answer: We appreciate the comments. BVDU (brivudine), we have edited in the revised manuscript.
10) Add references on the line 323.
Answer: We appreciate the comments. We have edited in the revised manuscript.
11) In the discussion, add some bibliographic information on XAV-939.
Answer: We appreciate the comments. We have edited in the revised manuscript.
12) TG 100-115 in Table 3 seems equally promising, in fact several scientific papers report its use in breast, liver and pancreatic cancer. I advise the authors to bring into the discussion the possible presence of molecules that are just as good as those chosen but excluded for a particular reason.
Answer: We appreciate the comments. We have edited in the revised manuscript.
13) Why compound J2 is not present in Figures 5 and 6 as BVDU? On line 108, the authors specify that for this work J2 and BVUD are used as control drugs.
Answer: We appreciate the comments. We continued with BVDU as it is the best choice based on the docking outcome.
14) Finally, I strongly advise the authors to recreate the tables and images following an alphabetical order of the components examined in order to facilitate the reader in searching and learning the text.
Answer: Thank you the comments. The authors believe that the manuscript data is very much discoverable comfortably in the current form.
Reviewer 3 Report
The present work discuss about the docking and molecular dynamic simulation for HSP27 inhibitors. This is is an interesting paper and focused mostly on insilico analysis of Hsp27. Few pointers
1. To my knowledge, current paper is focused on analysis for inhibition of Hsp27 are c-terminal domain inhibitors (PDB: 4MJH). I would request authors to clarify this in their manuscript and modify it with its implication on function of Hsp27.
2. Previous papers have shown classical phosphorylation inhibitions: https://www.jci.org/articles/view/130819
Ivermectin is the known drug. So please clarify how the identification of c-terminal inhibitors would help in filling the knowledge gap.
3. Introduction is too short and confusing. Why do you author thinks that small molecule inhibitor library is required for inhibiting Hsp27 from cancer or therapeutic perspective.
It does not explain how does Hsp27 functions if it is non ATP dependent chaperone. Also I am not sure if Hsp27 requires phosphorylation (line 65) for its activation then how it is ATP non-dependent molecular chaperone. Where is this phosphate group coming from? which kinase protein is phosphorylating it?
4. Authors are requested to provide image of mechanism of action for hsp27 and its inhibitory effect in figure1 instead of generalised graphical abstract of use of hsp27 as this is not a review paper. I doesn't go with theme of the work.
6. Can you please clarify if authors used similarity index for drug likeliness in comparison to known structures? (mentioned in subsection 2.2)
7. Did authors considered blind docking for their analysis? Can you please provide rationale behind selection of grid box for performing your analysis?
Author Response
#Reviewer 3:
The present work discusses about the docking and molecular dynamic simulation for HSP27 inhibitors. This is an interesting paper and focused mostly on in-silico analysis of Hsp27. Few pointers
- To my knowledge, current paper is focused on analysis for inhibition of Hsp27 are c-terminal domain inhibitors (PDB: 4MJH). I would request authors to clarify this in their manuscript and modify it with its implication on function of Hsp27.
Answer: We appreciate your comments. Point raised had been addressed appropriately in the manuscript.
- Previous papers have shown classical phosphorylation inhibitions: https://www.jci.org/articles/view/130819
Ivermectin is the known drug. So please clarify how the identification of c-terminal inhibitors would help in filling the knowledge gap.
Answer: We appreciate your comments. The interdimer interface formation involves the c-terminal, so a small molecule that can engage the amino acid residues involved in this process will be an advantage to the tackling of dimer formation. A point of note, these molecules used herein are already known for their anticancer aptitudes with different pathway of action.
- Introduction is too short and confusing. Why do you author thinks that small molecule inhibitor library is required for inhibiting Hsp27 from cancer or therapeutic perspective.
It does not explain how does Hsp27 functions if it is non ATP dependent chaperone. Also I am not sure if Hsp27 requires phosphorylation (line 65) for its activation then how it is ATP non-dependent molecular chaperone. Where is this phosphate group coming from? which kinase protein is phosphorylating it?
Answer: We appreciate your comments. The introduction has been improved as suggested. The small molecules inhibitor library considered in this current study were already synthesized and known anticancer drugs or molecules with different pathways. Getting a molecule from this library that can proceed to wet laboratory evaluations will be a plus to fight chemo-resistance in cancer.
- Authors are requested to provide image of mechanism of action for hsp27 and its inhibitory effect in figure1 instead of generalized graphical abstract of use of hsp27 as this is not a review paper. I don’t go with theme of the work.
Answer: We appreciate your comments. The figure in question is to show roles of HSP27 and not of mechanism of action. If replaced, it will defeat its purpose.
- Can you please clarify if authors used similarity index for drug likeliness in comparison to known structures? (mentioned in subsection 2.2)
Answer: We appreciate your comments. We did not use similarity index for drug likeliness.
- Did authors considered blind docking for their analysis? Can you please provide rationale behind selection of grid box for performing your analysis?
Answer: We appreciate your comments. The docking was not blind as we docked compounds to an established region on the protein that has been earlier reported to be made up of key amino acid residues involved in inter-dimer interactions.
Round 2
Reviewer 2 Report
The manuscript has improved in form but:
1) the list of references still does not follow the instructions requested by the journal. I invite the authors to read the Instructions for the authors.
2)Tables 1 and 2 should indicate S / N and return to recommend using alphabetical order.